# The Role of Community Pharmacists in the Detection of Clinically Relevant Drug-Related Problems in Chronic Kidney Disease Patients

**DOI:** 10.3390/pharmacy8020089

**Published:** 2020-05-22

**Authors:** Céline Mongaret, Léa Aubert, Amélie Lestrille, Victorine Albaut, Pierre Kreit, Emmanuelle Herlem, Natacha Noel, Fatouma Touré, François Lallier, Florian Slimano

**Affiliations:** 1Faculty of Pharmacy, Reims University, 51 rue Cognacq-Jay, 51100 Reims, France; laubert@chu-reims.fr (L.A.); amelie.lestrille@univ-reims.fr (A.L.); victorine.albaut@gmail.com (V.A.); 2Department of Pharmacy, CHU Reims, Avenue du Général Koenig, 51100 Reims, France; eherlem@chu-reims.fr; 3Pharmacie d’officine Croix du Sud, 13 avenue Léon Blum, 51100 Reims, France; 4Union Régionale des Professionnels de Santé (URPS) Pharmacien Grand Est, 18 quai Claude Le Lorrain, 54000 Nancy, France; pkreit@urpspharmaciensgrandest.fr; 5General practitioner office, 35 Place Luton, 51100 Reims, France; 6Department of Nephrology, CHU Reims, Avenue du Général Koenig, 51100 Reims, France; nnoel@chu-reims.fr (N.N.); ftoure@chu-reims.fr (F.T.); 7Faculty of Medicine, Reims University, 51 rue Cognacq-Jay, 51100 Reims, France; lallier.francois@orange.fr; 8General Practitioner Office, 15 Ter rue Charles de Gaulle, 51170 Ville-en-Tardenois, France

**Keywords:** chronic kidney disease, drug-related problems, community pharmacist, estimated glomerular filtration rate

## Abstract

Community pharmacists (CPs) have traditionally had limited access to patients’ estimated glomerular filtration rate (eGFR) during the medication-dispensing process. The increasing access to shared electronic health records is making eGFR available, but the skills needed to detect and manage clinically relevant drug-related problems (DRPs) are poorly documented. The primary objective of this study was to investigate the role of CPs in the medication-dispensation process for elderly patients with renal impairment. A total of 70 CPs participated in this 6 month study. Community pharmacists asked all patients ≥65 years to bring their laboratory test values for the next medication-dispensing process. Drug-related problem detection rates were compared between CPs (prospective period) and expert pharmacists (retrospectively). The clinical relevance of each DRP was assessed by nephrologists and general practitioners using an appropriate tool. Community pharmacists recruited n = 442 patients with eGFR < 60 mL/min/1.73 m^2^ and detected n = 99 DRPs, whereas expert pharmacists detected n = 184 DRPs. The most frequently detected DRPs were dosage problems and contraindications. According to assessment by clinicians, CPs and expert pharmacists identified 54.0% and 84.7% of clinically relevant DRPs, respectively. This study suggests a positive impact of the systematic availability of eGFR to CPs on the detection of several DRPs with clinical relevance.

## 1. Introduction

Chronic kidney disease (CKD) is an umbrella term for many conditions resulting in permanent kidney damage, which in turn can cause problems with medications. In nearly half of all cases, it develops as a complication of diabetes and hypertension [1], but these modifiable risk factors are only part of the problem. A progressive decline in renal function happens in normal ageing; starting at age 30, kidney function diminishes each year at an estimated rate of 1 mL/min, and even faster after age 65 [2,3]. Healthcare professionals should thus assume renal impairment when prescribing for the elderly [4]. Most drugs and metabolites are excreted by the kidneys; damage to these organs can lead to pharmacokinetic changes that can amplify the risk of adverse events and render elderly patients with CKD particularly prone to drug-related problems (DRPs).

Chronic kidney disease is a common health issue worldwide, affecting around 3 million people in France [5]. However, in many countries [6], including France [5], community pharmacists (CPs) cannot yet freely access laboratory data, including the estimated glomerular filtration rate (eGFR), which is currently the best available indicator of overall kidney function (and thus of renal impairment) [5].

Community pharmacists may help to detect DRPs when aware of their patients’ eGFR [7,8,9,10]. In a 2015 French study, the use of eGFR in 23 community pharmacies over 7 months led to the detection of 27 DRPs out of 1297 prescribed medicines (a DRP rate of 2.1%) [11].

In the era of shared electronic health records and the potential growing availability of eGFR to CPs, we wondered whether prescription problems in elderly patients with CKD could be avoided, and whether CPs have the required skills to detect DRPs and propose pharmacist interventions (PIs) required in cases of renal impairment.

This study investigated the importance of the role that CPs play in detecting prescription problems based on renal impairment (with free access to eGFR) and their ability to suggest appropriate PIs. The primary objective was to assess whether free access to eGFR would help community pharmacists to detect DRPs and adjust the medication-dispensing process for elderly patients (≥65 years old). We recorded CKD prevalence, DRPs/prescriptions, and the type of problem and drug involved. The secondary objective was to assess whether CPs have comparable skills to expert pharmacists in this domain. We compared their results with those of expert pharmacists in terms of number and type of DRPs, and the clinical relevance of interventions undertaken. Finally, we assessed community pharmacists’ satisfaction regarding the availability of eGFR to support dispensing medication.

## 2. Materials and Methods

### 2.1. Study Setting

This prospective non-randomized study was conducted in 70 community pharmacies that welcomed 70 pharmacy students (near graduation) for their last 6 month internship [12]. There was one pharmacy student per community pharmacy. Each community pharmacy had only one mentor pharmacist, agreed upon by the faculty of pharmacy sponsoring this study. All the mentor pharmacists had graduated at least five years ago. The community pharmacies were located in rural and urban areas, with regular or occasional patients. The study period ranged from January to the end of May 2017. All study material was collected by the beginning of June, even though the internship had finished at the end of the previous month.

### 2.2. Community Pharmacists’ Study Information

Academic pharmacists from the clinical pharmacy department of the faculty of pharmacy gave two study presentations to pharmacy students and their mentors (we refer to both herein as CPs). An e-learning resource was also sent to all pharmacy students and their mentors to help them train to score DRPs and apply standard-of-care PIs according to the French Society of Clinical Pharmacy [13,14]. There was no other training. Finally, we gave each community pharmacist a form to contact the clinical pharmacy department sponsoring the study in case any further questions arose.

### 2.3. Inclusion Criteria

We gave written information concerning the study’s objectives to each prospective study participant and assessed their eligibility based on the following criteria: patients had to have eGFR data indicating kidney damage (with the abbreviated modification of diet in renal disease (aMDRD) [15] or a chronic kidney disease—epidemiology collaboration equation (CKD-EPI) of < 60 mL/min/1.73 m^2^) [16]; to be on at least one prescribed medication; to be able to understand the study and its objectives; and to agree to participate. Verbal informed consent was obtained from all participants.

### 2.4. Exclusion Criteria

We excluded patients from the study if they were younger than 65 years old; had an eGFR that did not indicate kidney damage (≥60 mL/min/1.73m^2^) or that dated from more than a year ago; were currently undergoing hemodialysis (because we supposed that this population was under the care of hemodialysis centers where nephrologists were already providing careful monitoring); or were unable to understand the study.

### 2.5. Study Design

The steps of the design are described in Figure 1.

### 2.6. Month Internship Period

Community pharmacists pre-selected potential study participants based on incomplete inclusion criteria. To obtain their eGFR values and confirm inclusion, they asked selected patients to bring their laboratory test results with them on their next visit to the pharmacy.

From this point, community pharmacists accounted for eGFR while dispensing medications: if they detected a potential DRP, they had to assess it, record it, and suggest (to the patient’s prescriber, typically his/her general practitioner) an appropriate PI using the specific tools of the French Society of Clinical Pharmacy [13,14]. The interventions could be accepted or refused by the practitioner, with or without justification [13].

Each patient was given an anonymous patient record used to collect data concerning age, prescription, laboratory test values (restricted to eGFR), DRP(s) detected, and PIs suggested by CPs. We also collected data on time spent for each patient’s record (less than 5 min, 5–10 min, 10–20 min, 20–30 min, and more than 30 min), including prescriber contact. At the end of the internship period, all the patient records were finally centralized by the clinical pharmacy department for further analyses.

### 2.7. Retrospective Dispensing Process by Expert Pharmacists and Comparison with Community Pharmacists

After the end of Phase 1 (prospective) and the centralization of all patient records, two of the four clinical pharmacists (the expert pharmacists) from the promoting academic department reviewed all the records of patient who met the inclusion criteria. Half of these experts were community and academic pharmacists, half were hospital and academic pharmacists, and all had training in clinical pharmacy. Each patient record was retrospectively and independently reviewed. Each DRP related to eGFR detected as classified according to the French Society of Clinical Pharmacy tool [13]. All interventions were theoretical because of their retrospective nature, and were also classified in accordance to the SFPC tool. In case of disagreement, a third expert pharmacist opinion was added to reach a consensus.

### 2.8. Comparative Clinical Assessment of Pharmacists’ DRPs

Three clinicians (one hospital nephrologist and two general practitioners (GPs), including one with pharmacovigilance experience) independently assessed whether PIs were clinically relevant, using a scale (modified Chedru tool [17,18]) ranging from −1, for interventions that seemed to be detrimental to the patient, to 3, for interventions considered to have averted potentially fatal accidents (Appendix A). An impact of ≥1 was defined as clinically significant. They reviewed the clinical relevance of all DRPs and interventions performed by both community and expert pharmacists. We finally compared the clinical assessment between both groups (see Section 2.10).

### 2.9. Satisfaction Form

After the study, we asked community pharmacists to fill in a satisfaction form (using a Likert scale) about how useful they found the systematic availability of eGFR (from “very useful” to “very useless”) and how easy they found performing the interventions (from “very easy” to “very hard”). We also asked questions on their assessment of the quality of communication between themselves and the prescriber.

### 2.10. Statistical Analysis

All data were collected in an Excel^®^ spreadsheet (Microsoft Corporation, Redmond, WA, USA). Agreement between the community and expert pharmacists about the detection and classification of DRP was assessed by describing the proportion of cases with observed agreement and estimating the kappa statistic with a 95% confidence interval [19,20]. Confidence intervals containing 0 were interpreted as not significant. The kappa statistic was represented as a fraction (i.e., actual agreement beyond chance/potential agreement beyond chance) and fell between −1 and 1. Kappa statistics were interpreted according to common guidelines as poor (0), slight (0–0.2), fair (0.2–0.4), moderate (0.4–0.6), substantial (0.6–0.8), or almost perfect (0.8–1). All data were computed using R^®^ (The R Project for Statistical Computing, v.3.2.2). The significance level for the first alpha type was 0.05.

### 2.11. Ethics Approval

The medicines, renal impairment, and community pharmacists (Médicaments, Insuffisance Rénale et Pharmacien d’Officine, MIRPhO) were registered by the University of Reims and the study was reviewed and approved by the Area of Ethical Reflection of Champagne-Ardenne.

## 3. Results

We found a prevalence of CKD (eGFR less than 60 mL/min/1.73 m^2^) of 21.5% in patients older than 65, in accordance with other studies conducted in similar settings [11,21,22]. The prevalence of CKD strongly depends on the equations used to calculate eGFR. We chose, and recommended that CPs also choose, aMDRD and CKD-EPI (and not Cockcroft–Gault), because these equations show greater consistency [21,23].

### 3.1. Population

A total of 2055 patients from 70 pharmacies agreed to bring their laboratory results with them on their next visit. Of these, 482 (23.5%) had an eGFR indicating kidney disease, and we included 442 (21.6% of the total number) for further analysis (Figure 2).

A total of 167 males (38.2%) and 275 females (61.8%) took part in the study. Their mean age (±standard deviation) was 81.5 ± 6.6 years old and most (77.4%) were 75 or older. A total of 4508 prescribed medicines were reviewed (10.2 ± 4.5 per patient) (Appendix A). We classified patients based on the stage of their kidney disease and found that 262 (59.3% of the 442 included) patients were at Stage 3A, 124 (28.1%) at Stage 3B, and 56 (12.7%) at Stage 4 or 5. The distribution in stages did not seem to depend on age or sex, and both formulas used to calculate eGFR, aMDRD and CKD-EPI, yielded similar results. In terms of patients prescribed medicine, there were no differences based on sex (10.5 ± 4.2 for males and 10.1 ± 4.7 for females) or whether the patient was over 75 years old (10.3 ± 4.6) or younger (10.2 ± 4.0). The number of prescribed medicines according to the stage of renal impairment was 9.5 ± 4.4, 10.9 ± 4.2, and 12.0 ± 4.7 for Stage 3A, Stage 3B, and Stage 4/5, respectively. Only 9.1% of patients were prescribed fewer than five medicines.

### 3.2. Population DRP Detection by Community Pharmacists and Expert Pharmacists

We compared the decisions made by CPs with those made by expert pharmacists concerning detection and classification of (categories of and drugs involved in) DRPs. Expert pharmacists reviewed all patients’ files (n = 442) and detected 184 DRPs (4.1%) among 4508 medicines, whereas community pharmacists detected n = 99, also in 442 patients (2.2%). More than 70% of all DRPs were due to drug dosage. Unlike experts, CPs did not record any DRPs due to monitoring, therapeutic inconsistency, or improper prescriptions (lack of information or clarity) (Table 1).

Although both expert and community pharmacists agreed that antidiabetics were commonly involved, they differed in opinion otherwise and, strikingly so, regarding angiotensin-converting enzyme (ACE) inhibitors, diuretics, and analgesics. Community pharmacists found ACE inhibitors second to antidiabetics in involvement in DRPs, whereas expert pharmacists reported very few DRPs in this class. Concerning diuretics, CPs found 15 DRPs and experts found none. Expert pharmacists found that analgesics were involved in one in four DRPs (26.6%), with acetaminophen (14.7%) and non-steroidal anti-inflammatory drugs (NSAIDs) (9.2%) accounting together for almost all problems in the class. In this class, experts found about 10 times the number of DRPs than CPs did.

Community pharmacists suggested interventions (and contacted prescribers) in 7 out of 10 identified DRPs (67.7%). Most of the time, prescribers reviewed their prescriptions based on pharmacists’ comments (52.5% of the time). Community and expert pharmacists found 42 common DRPs. Community pharmacists reported 54 DRPs that experts did not report. Expert pharmacists reported 142 DRPs that community pharmacists did not report.

There was strong agreement between CPs and expert pharmacists about the DRP category “no DRPs”, and agreement about the DRP categories “contraindication/non-conformity to guidelines” and “dosage problems”. We could find no statistically significant agreement in any other categories (higher than the expected agreement, with a kappa statistic k = 0.168).

The procedure somewhat prolonged dispensing time, but one in four (25.9%) remained shorter than 5 min and only about 3% (3.2%) took longer than 30 min.

### 3.3. Comparative Clinical Relevance Assessment of DRPs

Three clinicians (a nephrologist and two general practitioners) independently classified DRPs by clinical relevance. Table 2 compares community and expert pharmacists’ decisions; DRPs spotted by experts, no matter which expert, were significantly (*p* < 0.001) more relevant than those spotted by CPs.

The clinicians rated interventions by community pharmacists and expert pharmacists. Interventions rated “−1” (for potentially harmful interventions) exclusively concerned ACE inhibitors and were mainly suggested by community pharmacists (Appendix A). 

### 3.4. Satisfaction Form

A total of 52 community pharmacists (74.2%) completed the post-study satisfaction form. For one in three (36.5%), the availability of eGFR significantly improved dispensing and most (n = 39, 75.0%) reported having been sufficiently prepared for this practice during their pharmaceutical education. About half of them deemed communication with physicians to be quite difficult (n = 25, 48.1%).

## 4. Discussion

With a prevalence of 20.1%, CKD was as common as expected for elderly patients in this study setting [10,20,21]. We found that free access to eGFR allowed community pharmacists to detect and perform pharmacist interventions for 99 DRPs for 442 patients of this population (≥65 years old).

Prescribers, typically the patient’s GP, accepted half (52.5%) of the PIs suggested by CPs. In a French pilot study by Pourrat et al. the acceptance rate of pharmacist interventions was 33.3%, but the CPs only reported 18 interventions [11]. In our study, only 66 out of 99 interventions performed by CPs pharmacists were sent to prescribers due to lack of GP availability; this could perhaps explain this difference.

On both sides, efforts are needed to improve communication between pharmacists and doctors. A total of 48% of CPs expressed difficulties with this. We believe this issue to be an obstacle to a potential reduction of DRPs. In our study, not only prevalence but also the type of DRP indicated that clinicians, notably general practitioners, may lack knowledge of how to adapt some medication prescriptions to eGFR. Sound education of both pharmacists and prescribers is needed, and modifications in prescriptions should be easy to discuss.

Several medications need to be adjusted or avoided depending on the stage of CKD. We confirmed [12,24] dosage issues (74% for CPs, 72% for expert pharmacists) and contraindications/non-conformity to guidelines (26% for CPs, and 14% for expert pharmacists) as the commonest types of DRP. Moreover, results from the chronic kidney disease—renal epidemiology and information network (CKD-REIN) cohort showed a smaller estimation. We believe that this can partly be explained by the fact that in the CKD-REIN cohort, only experts and selected patients reviewed medical reports. Still, in their study, dosage problems (35%) and contraindications (31%) also represented the most common categories [24]. The detection of other DRP categories such as “untreated medical indication” was not found in our study. Indeed, the systematic availability of eGFR has been suggested to encourage detection of DRPs related to the renal safety of prescribed medicines instead of DRPs related to the specific CKD condition. Even if we did not collect comorbidities among the 442 patients recruited, we hypothesize that not all patients suffered from CKD. A large panel of DRP categories is more probable with a different design, for example, by focusing on CKD patients in an ambulatory setting such as day hospital or a nephrologist conventional care consultation. For example, such a design was performed by Belaiche et al. in a 6 month study targeting DRPs in CKD patients [25]. They found n = 142 DRPs among 67 successive outpatients, where the most frequent DRP was untreated medical indication (31.7%) such as iron deficiency, found in more than two-thirds of cases. This kind of DRP detection requires access to biological values other than eGFR, which was not the case in our study. The other most frequent DRPs were dosage problems (19.0%) and contraindication/non-conformity to guidelines (12.0%). Antidiabetics are of particular concern: we confirmed [24] that oral antidiabetics were one of the pharmacological classes most frequently involved in DRPs (20.2% for community and 21.7% for expert pharmacists). Our results were concordant with the CKD-REIN cohort, where 20.8% of patients had at least one inappropriate prescription of antidiabetics [24]. Kidney damage, one of the long-term complications of diabetes, occurs in approximately 40% of diabetic patients [26], but treating both diseases can be complicated; several medications for diabetes have dosage problems or contraindications regarding eGFR [27]. For example, metformin, one of the most prescribed antidiabetics, must be adapted to eGFR to avoid a rare but life-threatening complication called lactic acidosis [28]. These issues have to be addressed at a prescription level and, if data on eGFR become available, at a dispensing level also.

However, accessing biological data is not quite enough to prevent DRP occurrence. In the existing literature, DRPs related to eGFR vary widely, from 13% to 53% [23,29,30], reflecting variations in study design, cultural aspects, and individuals. In addition, we are inclined to believe that training and experience (particularly in a hospital setting) are the main factors influencing DRP detection [31]. We found important variations in the prevalence of DRPs when assessed by expert (41.6%) versus community (22.4%) pharmacists. The expert pharmacists of our study spotted almost twice the number (1.9-fold) of DRPs that community pharmacists spotted. This gap in DRP detection confirms the results of a pilot study where expert pharmacists identified 2.1 times the number community pharmacists did [11].

One of the most striking differences between expert and community pharmacists concerned analgesics, which were involved in 5.05% of DRPs for community pharmacist assessments versus 26.6% of DRPs for expert pharmacists. The expert pharmacists’ findings were similar to those from the CKD-REIN cohort, where 20.9% of DRPs involved analgesics [24]. We believe that experts identified DRPs better because they were more vigilant concerning this class of medication. NSAIDs are very common and can easily be overlooked. Moreover, our study only reviewed prescribed medicines, and we could hypothesize that this DRP rate was underestimated in accordance with our study design.

All health professionals should be educated on interactions and problems with dosage in analgesics; they can be detrimental to renal function [32,33,34], especially NSAIDs, and they are not only widely prescribed but also available as over-the-counter medications (OTCs). This is also true for acetaminophen, for which a dose adjustment is needed when eGFR < 50 mL/min/1.73 m^2^ [35,36].

Our study showed that DRPs detected by community pharmacists were significantly less relevant than DRPs detected by expert pharmacists. Our three independent reviewers (a nephrologist and two general practitioners) assessed, for example, all interventions concerning ACE inhibitors as detrimental. Even though ACE inhibitors are mainly eliminated by renal excretion [37,38], they have an indirect protective effect by reducing proteinuria and exerting antihypertensive properties [39]. Pharmacists should restrict their interventions only to patients taking them with NSAIDs [40]. In our study, we suggest that community pharmacists did not seem to be aware of this, as they reported almost all DRPs associated with ACE inhibitors (17% of DRPs detected vs. 2.2% for experts). 

Taken together, our key findings demonstrate that community pharmacists are able to detect some DRPs based on eGFR as such, but they also show that training probably improves DRP detection rate and intervention safety. There is an important need for education (of both CPs and GPs) in detecting DRPs and in tailoring medication prescriptions to eGFR. With the at-risk population always increasing, particular attention has to be paid to medications for diabetes and analgesics (and especially OTC medications). Better communication between pharmacists and prescribers would also improve prescription safety and decrease the rate of DRPs.

This study had some limitations. First, we did not record co-morbidities of patients or other laboratory test values, which could influence the rate or type of DRPs. However, the aim of the study was not to explore clinical or biological risk factors, and it would have complicated the statistical analysis without benefit. Secondly, the study did not assess whether DRP detection by CPs improved during the 6 month period. It is known that time and habit improve intervention quality in healthcare [41]. Thirdly, we also considered a possible selection bias: patients who agree to bring their laboratory tests back may be more careful about medication problems. Finally, CPs checked for DRPs only on prescribed medications. This could be a bias especially towards analgesics, such as acetaminophen and NSAIDs. Other studies on this topic have reported the same limitations [11,24].

## 5. Conclusions

The number of prescription problems in elderly patients with CKD could be decreased with CP access to eGFR. This study showed that they are able to detect and manage some clinically relevant DRPs. However, training and experience have a big influence on quality of DRP detection and on the relevance of PIs, and emphasis on this subject in university education could help to decrease remaining gaps in knowledge. Better education of doctors and pharmacists, free access to eGFR for community pharmacists, and improved collaboration between health professionals all play a role in decreasing DRPs linked to eGFR in elderly patients with CKD. Further studies will have to confirm our results, first in a context of experimentation and then related to national health authorities.

## Figures and Tables

**Figure 1 pharmacy-08-00089-f001:**
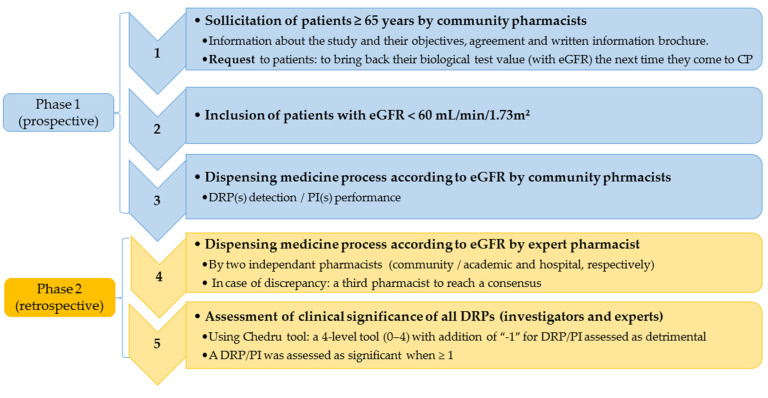
Study design. The three first steps were the prospective phase of the study. The last two steps were retrospective and were performed by expert pharmacists and clinicians, respectively. eGFR: estimated glomerular filtration rate; DRP: drug-related problem; PI: pharmacist intervention.

**Figure 2 pharmacy-08-00089-f002:**
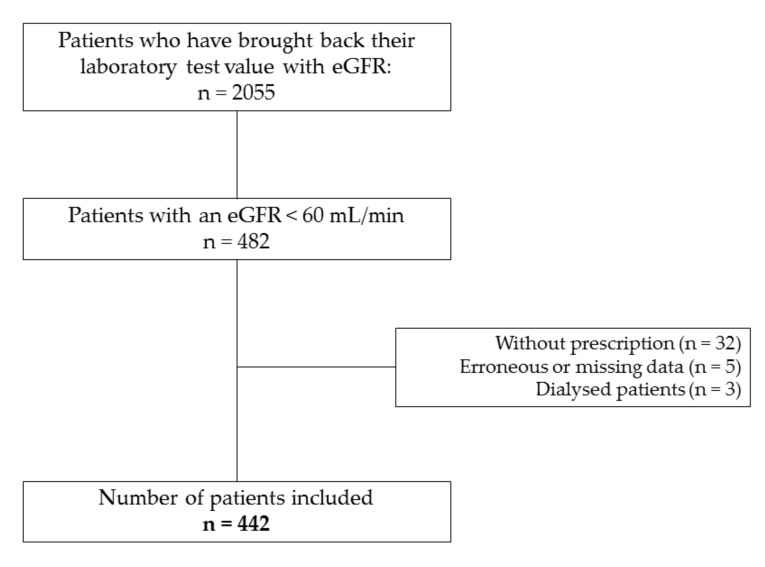
Flow chart of the study.

**Table 1 pharmacy-08-00089-t001:** Number and classification of DRPs performed by community and expert pharmacists.

	Community Pharmacists n (%)	Expert Pharmacists n (%)
Number of DRPs (categories below)	99	184
Dosage problem	73 (73.7)	133 (72.3)
Contraindication/non-conformity to guidelines	26 (26.3)	26 (14.1)
Monitoring	-	14 (7.6)
Therapeutic redundancy	-	5 (2.7)
Improper prescription: lack of information, of clarity	-	6 (3.3)
Drug Classes		
Antidiabetics	20 (20.3)	40 (21.7)
Anti-gout preparations	12 (12.1)	29 (15.8)
Analgesics	5 (5.1)	49 (26.6)
ACE inhibitors	17 (17.2)	4 (2.2)
Diuretics	15 (15.2)	-
Others	30 (30.3)	62 (33.7)
Number of Interventions (Categories Below)	99	184 *
Dose adjustment	72 (72.7)	126 (68.5)
Drug switch/establishment of a therapeutic alternative	12 (12.1)	29 (15.8)
Drug monitoring	9 (9.1)	18 (9.8)
Discontinuation or refusal to deliver	6 (6.1)	5 (2.7)
Optimization of the dispensing/administration modalities	-	6 (3.3)

ACE: angiotensin-converting enzyme; other drug classes included lipid-modifying agents, psycholeptics, antiepileptics, cardiac therapy, systemic antibacterial (for systemic use), and antihistamines; * these interventions were proposed but not performed (retrospective analysis).

**Table 2 pharmacy-08-00089-t002:** Clinical relevance of DRPs performed by community and expert pharmacists. Clinical relevance was assessed using the modified Chedru tool.

	Community Pharmacists’ DRPs				Expert Pharmacists’ DRPs			
Clinical relevance	Nephrologist	General practitioner 1	General practitioner 2	*p*	Nephrologist	General practitioner 1	General practitioner 2	*p*
0	39 (49.4)	38 (46.9)	32 (41.2)	NS	26 (17.8)	37 (25.2)	8 (5.4)	<0.001
≥1	40 (50.6)	43 (53.1)	46 (57.8)		120 (82.2)	110 (74.8)	142 (94.6)	

NS: non-significant.

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
