# Peer review of "The Role of Community Pharmacists in the Detection of Clinically Relevant Drug-Related Problems in Chronic Kidney Disease Patients"

_pharmacy, 2020, doi:10.3390/pharmacy8020089_

Round 1
Reviewer 1 Report
Overall this is an interesting paper and good idea to study the community pharmacists involvement in CKD patients.
The paper would benefit from a medical writer- I appreciate that English is not the primary langauage of the investigators - however, there are many grammatical errors.
Some issues and consideration for improvement on study design:
Overall- interesting findings but more details in methods and results are required:
1) Info on the pharmacists- years of practice type of pharmacy
2) More info on the pts- number of meds- type of meds, comordidities- cannot determine DTPS without this type of information
3)Info on the types of interventions..analgesics and ACEin- want more info. What were the interventions?
4) Table 3- unclear as to what was considered relevant…need more info- such as how did the pharmacist rank them?? And were the nephrologists and general practitioners the same for both cohorts? Who was compared- meaning between just community pharmacists and specific practitioners. What about the DTPs they missed and vice versa? Which DTPs? The ones that were intervened or all the potential ones?
5) Methods- very unclear on experts role- did they review all 482 pts with decreased eGFR and all their meds? And not sure how you can determine DTP without all the info on the patients. As for the results of Table 3- how was this done- not clear in your methods.
I enjoyed reading the paper but would definitely like more details in the methods and results as stated above
Author Response
Note: Our answers to the Reviewers’ comments are written in blue font below, and the modifications are highlighted in blue in the revised version of the manuscript.
Overall this is an interesting paper and good idea to study the community pharmacists involvement in CKD patients.
The paper would benefit from a medical writer- I appreciate that English is not the primary language of the investigators - however, there are many grammatical errors.
We would like to thank you for your interest in our manuscript. In order to improve English of the manuscript it has been proofread by a medical writer (English Editing MDPI service).
Some issues and consideration for improvement on study design:
Overall- interesting findings but more details in methods and results are required:
- Info on the pharmacists- years of practice type of pharmacy
All participating community pharmacies were those who were allowed to welcome a near-graduated pharmacy student. In France, mentor community pharmacists need to be at least 5-years post-graduated with regular training.
The community pharmacies were distributed between rural and urban areas with regular or occasional patients. We did not include this distribution into the Results analysis. Indeed most of the community pharmacies in the most populous city of the study (200,000 inhabitants) have been with regular patients and this criterion appeared to be irrelevant in our context.
We complete the first paragraph in the Method section as follow (Lines 71-75): “There was one pharmacy student per community pharmacy. Each community pharmacy had only one mentor pharmacist agreed by the faculty of pharmacy sponsoring this study. All the mentor pharmacists had graduated at least five years ago. The community pharmacies were located in rural and urban areas, with regular or occasional patients."
- More info on the pts- number of meds- type of meds, comordidities- cannot determine DTPS without this type of information
For each patient included the following data were collected: age, gender, eGFR, number and type of medicines prescribed. We only analysed the data for the 442 patients with eGFR<60mL/min. They had (mean±Standard Deviation) 10.2±4.5 prescribed medicines. According to gender, there were 10.5 ± 4.2 medicines for male and 10.1±4.7 for female. Patients over 75 years-old received 10.3±4.6 and patients less than 75 yo 10.2±4.0. Number of prescribed medicines according to the stage of renal impairment was: 9.5±4.4, 10.9±4.2 and 12.0±4.7 for stage 3A, stage 3B and stage 4/5, respectively. Only 9.1% of patients have less than 5 prescribed medicines. We added the information above into the appropriate Result section (lines 178-182) as follow: “there were no differences based on gender (10.5 ± 4.2 for male and 10.1±4.7 for female) or whether the patients were over 75 years old (10.3±4.6) or under (10.2±4.0). The number of prescribed medicines according to the stage of renal impairment was 9.5±4.4, 10.9±4.2, and 12.0±4.7 for stage 3A, stage 3B, and stage 4/5, respectively. Only 9.1% of patients were prescribed fewer than 5 medicines.”
According to the ATC classification (WHO-Collaborating Centre for Drug Statistics Methodology) the third most prescribed medicines were “Cardiovascular system” (26.8%), “Alimentary tract and metabolism” (19.3%) and “Nervous system” (13.5%), respectively. Please find below a Table (added as the Table S3) with the ten most prescribed medicines subclasses of the ATC classification:
Table 1: Most frequently prescribed medicines (n = 4,508 prescribed medicines)
|
Drug groups (ATC classification system) |
n (%) |
|
C - Cardiovascular system |
1210 (26.8) |
|
Diuretics (C03) |
303 (6.7) |
|
Renin-angiotensin system blockers (C09) |
229 (5.1) |
|
Hypolipemic drugs (C10) |
196 (4.3) |
|
Beta blocking agents (C07) |
185 (4.1) |
|
A - Alimentary tract and metabolism |
871 (19.3) |
|
Drugs for peptic ulcer and reflux disease (A02 ) |
210 (4.7) |
|
Drugs used for diabetes (A10) |
207 (4.6) |
|
N – Nervous system |
609 (13.5) |
|
Analgesics (N02) |
267 (5.9) |
|
Psycholeptics (N05) |
160 (3.5) |
|
B – Blood and blood-forming organs |
306 (6.8) |
|
Antithrombotics drugs (B01) |
260 (5.8) |
|
R- Respiratory system |
262 (5.8) |
|
Drugs for obstructive airway diseases (R03) |
150 (3.3) |
Regarding the patients details, we did not design our study to collect comorbidities and we already added this as a limit at the end of our manuscript. We agree with Reviewer #1 that comorbidities knowledges could help to improve pharmacist intervention in patients with CKD. We hypothesize that most of the patients has hypertension and/or diabetes mellitus [1].
[1] Couser WG, Remuzzi G, Mendis S, Tonelli M. The contribution of chronic kidney disease to the global burden of major noncommunicable diseases. Kidney Int 2011; 80: 1258–70.
- Info on the types of interventions..analgesics and ACEin- want more info. What were the interventions?
We have completed the Table 1 (line 189) with the following information:
Among the n=99 DRPs detected by Community Pharmacists, there was:
- N=72 Dose adjustment
- N=9 Drug monitoring
- N=12 Drug switch
- N=6 Drug discontinuation
Among the n=184 DRPs retrospectively detected by Expert Pharmacists, there was:
- N=126 Dose adjustments
- N=6 Administration mode optimisation
- N=5 Drug discontinuation
- N=29 Drug switch
- N=18 Drug monitoring
There was n=17 community pharmacist interventions on ACEi distributed as follow:
- N=14 Drug adjustment
- N=2 Drug switch
- N=1 Drug monitoring
The four interventions by Expert pharmacists for ACEi were Drug adjustment.
For analgesics such as acetaminophen, several interventions were Drug adjustment for patients with eGFR between 15-50mL/min (maximum 3g daily) or eGFR<60mL/min (maximum 2g daily). For NSAIDs the main interventions proposed by Expert pharmacists was Drug discontinuation.
All the interventions described were according to the tool published by Vo, T.H.;, et al. J. Clin. Pharm. Ther. 2018; 43: 240-248. doi: 10.1111/jcpt.12642).
- Table 3- unclear as to what was considered relevant…need more info- such as how did the pharmacist rank them?? And were the nephrologists and general practitioners the same for both cohorts? Who was compared- meaning between just community pharmacists and specific practitioners. What about the DTPs they missed and vice versa? Which DTPs? The ones that were intervened or all the potential ones?
As described in the Method section, the clinical relevance has been assessed according to the Chedru tool (please see Supplementary Table S1). In order to improve the reporting of Results, we assume that a pharmacist intervention was significant as soon as the clinical relevance was positive, even if minor (≥1). This is reports in the Method section as follow: “An impact ≥ 1 was defined as clinically significant” (line 133).
The clinicians who rated the pharmacists interventions were the same (nephrologists and general practitioners) when rating community as well as expert interventions. This is reports in the Method section as follow: “They reviewed the clinical relevance of all DRPs and interventions performed by both community and expert pharmacists. We finally compared the clinical assessment between both groups” (lines 133-135).
However we did not performed a comparison intragroup for Community pharmacists or for Experts pharmacists.
Finally there were n=42 common DRPs between Community and Expert Pharmacists. Community Pharmacists reported n=54 DRPs that Experts pharmacists did not report while Expert pharmacists reported n=142 DRPs whereas Community pharmacists did not report.
However we did not performed the analysis regarding the kind of DRP/interventions reported according to the pharmacist group. We also did not performed the analysis regarding the DRP/interventions missed.
- Methods- very unclear on experts role- did they review all 482 pts with decreased eGFR and all their meds? And not sure how you can determine DTP without all the info on the patients. As for the results of Table 3- how was this done- not clear in your methods.
As we specify in the Method section, the Expert pharmacists retrospectively reviewed the prescription from the n=442 patients who met the inclusion criteria (stage 4 phase 2 in the Figure 1). They were in the same conditions as for Community pharmacists except for the patient contact. We agree with reviewer that it is difficult to formally state on a DRP (DTP) without all the info on the patients but the study has been designed to reproduce the real life condition of community pharmacy practice and they usually have limited access to relevant information (clinical conditions, biological values as eGFR, …).
We rephrased this part in the Method section in order to improve the understanding for readers: “After the end of the phase 1 (prospective) and the centralization of all patient records, two of the four clinical pharmacists (the Expert pharmacists) from the promoting academic department reviewed all the patient records who met the inclusion criteria. Half Expert were community and academic pharmacists, half were hospital and academic pharmacists, all with training in clinical pharmacy. Each patient record was retrospectively and independently reviewed. Each DRP related to eGFR detected has been classified according to the French Society of Clinical Pharmacy’s tool. All interventions were theoretical because retrospective, and also classified in accordance to the SFPC tool. In case of disagreement, a third expert pharmacist opinion served to reach a consensus” (lines 122-129).
I enjoyed reading the paper but would definitely like more details in the methods and results as stated above
Thank you very much for helping us to improve our manuscript.
Reviewer 2 Report
Even though this paper is adding a value to the current literature, it needs further minor revision. More specific comments are following:
Introduction:
Line 36: Please add abbreviation for chronic kidney disease
Line 44: Please delete “referred to as” so, it can be abbreviated as “DRPs”
Line 49: Use the abbreviation instead of defining it again
Line 50: Could you add a sentence explain the significance or take-away message from this study? Is 27 DRP too many or too little in 23 community pharmacies?
Line 57-58: This sentence is confusing, please rephrase it.
Line 60: Use the abbreviation instead of defining it again
Line 61: I think you meant, “CKD” not “CDK”
Materials and Methods:
Study setting: how did pharmacy students get involved in this study? Did they collect the data after pharmacists’ interventions?
Line 76-77: Is this a training provided by pharmacist faculty to students and community pharmacists? If so, how did students get involved in this study? Also, did you do this training in 70 community pharmacies?
Line 84: eGFR is already abbreviated on line 47
Line 85 and 86: Please provide references for aMDRD and CKD-EPI
Figure 1: Please delete “a” in front of eGFR<60ml/min. Also, the unit for eGFR should be mL/min/1.73m2
Line 134: Was this satisfaction form in a Likert scale?
Results:
Section 3.2: how many prescriptions were reviewed from 442 patients? This may be relevant information to determine the DRP prevalence in elderly patients with renal impairment. Also, were community pharmacists trained on DRP categories? How many community pharmacists were involved in this study from 70 community pharmacies?
Line 209: does 30 minutes including waiting for MD’s response after leaving the message?
Table 1:
Do you have any patient characteristics? For example, was DRP more common in advanced CKD stage? Or patients who are older than 70? Also, what do you mean by anti-gout “preparations”?
How was five categories of DRP chosen? Or were you only able to determine five different categories within DRP? For instance, you did not see any DRP in failure to receive drug nor drug interaction.
Please consider subcategorizing dosage problems into overdose and subtherapeutic dosage. This will be critical because majority (70%) of DRP is under dosage problems. Expert pharmacists found 4 DRP related to ACE inhibitors when 17 DRP were detected by CP. Of which, some intervention related to ACE inhibitors were considered as harmful to patients. By subcategorizing the dosage problem, it may explain the significance of DRPs.
Table 2:
I am not sure if this table adds value in this manuscript with these reasons:
- There were only 42 common DRP reported by community pharmacists and expert pharmacists.
- Thus, you will see strongly agreed on no DRP (because there aren’t many DRP)
- Some legends are not related to the table such as a definition of SE.
- Investigator should be replaced with community pharmacists to be consistent with Table 1
Discussion:
- Line 229: I believe it should be greater or equal to 65 years old in the bracket.
- Line 232-234: this sentence “it is always possible…” does not make sense.
- Line 276: I am confused. Analgesic DRP that was detected by experts (27%) were OTC drugs (NSAIDs)? If so, how did expert pharmacists were able to detect this DRP retrospectively when community pharmacists could not?
- Line 287: what type of dosage DRP did community pharmacist say about ACE inhibitor? Too high or too low? How would you know if the CPs did not know ACE inhibitors were used for renal protection?
- Line 291: what information makes you conclude that the training greatly improved DRP detection rate and intervention safety?
- It would be critical to discuss how your study was not able to detect DRP categories such as adverse drug events and untreated indication.
References:
All references’ numbers are duplicated.
Author Response
Note: Our answers to the Reviewers’ comments are written in blue font below, and the modifications are highlighted in blue in the revised version of the manuscript.
Reviewer #2
Even though this paper is adding a value to the current literature, it needs further minor revision. More specific comments are following:
Thank you for the feedback and your help for improving our manuscript.
Introduction:
Line 36: Please add abbreviation for chronic kidney disease
This is added as follow: “Chronic kidney disease (CKD) is an umbrella term for many conditions resulting in permanent kidney damage, which in turn can cause problems with medications”. (lines 38-39)
Line 44: Please delete “referred to as” so, it can be abbreviated as “DRPs”
This is removed as follow: “[…] particularly prone to drug-related problems (DRPs)”. (line 46)
Line 49: Use the abbreviation instead of defining it again
This is corrected as follow: “Community pharmacists may help detect DRPs when […]” (line 52)
Line 50: Could you add a sentence explain the significance or take-away message from this study? Is 27 DRP too many or too little in 23 community pharmacies?
To rephrase, there was not n=1297 prescriptions but there was n=1297 prescribed medicines in the study by Pourrat et al. (this is now corrected), so a rate of DRP (related to eGFR) detected of 2.1%. This study was the first in our country to investigate this issue and we cannot compare to other studies with a similar design. However, in our study, there was n=99 DRPs detected out of 4508 prescribed medicines, and the rate of detected DRP was 2.2%. We cannot assess if this is too many or too little but there seem to be a consistence between both studies. We rephrased as follow: “In a 2015 French study, the use of eGFR in 23 community pharmacies over 7 months led to the detection of 27 DRPs out of 1297 prescribed medicines (a DRP rate of 2.1%)” (lines 53-54).
Line 57-58: This sentence is confusing, please rephrase it.
We rephrased as follow: “This study investigates the importance of the role that CPs play in detecting prescription problems based on renal impairment (with free access to eGFR) and their ability in suggesting appropriate PIs” (lines 59-61).
Line 60: Use the abbreviation instead of defining it again
This is corrected as follow: “The primary objective was to assess whether free access to eGFR would help community pharmacists detect DRPs and adjust the medication-dispensing process for elderly patients (≥65 years old)” (lines 61-63).
Line 61: I think you meant, “CKD” not “CDK”
You are right, this is corrected (line 63).
Materials and Methods:
Study setting: how did pharmacy students get involved in this study? Did they collect the data after pharmacists’ interventions?
This prospective non-randomized study was conducted in the 70 community pharmacies that welcomed a pharmacy student (from the faculty of pharmacy promoting the study) during their last 6-months internship before graduation. All pharmacy students were involved and the achievement of this project was included in their final evaluation. This prospective part corresponds to the phase 1 (Figure 1). All pharmacy students and their mentors performed the study in each community pharmacy and, at the end of the prospective phase, all the patients record collected by the pharmacy student were centralized in the faculty of pharmacy. We completed the Method section to clarify for readers: “This prospective non-randomized study was conducted in 70 community pharmacies that welcomed 70 pharmacy students (near graduation) for their last six-month internship. There was one pharmacy student per community pharmacy. Each community pharmacy had only one mentor pharmacist agreed by the faculty of pharmacy sponsoring this study. All the mentor pharmacists had graduated at least five years ago. The community pharmacies were located in rural and urban areas, with regular or occasional patients” (lines 70-75).
Line 76-77: Is this a training provided by pharmacist faculty to students and community pharmacists? If so, how did students get involved in this study? Also, did you do this training in 70 community pharmacies?
There was no specific training by the academic pharmacists to the community pharmacists except:
- Two presentations study for pharmacy students and their mentor (in the faculty of pharmacy at a dedicated time frame) dedicated to the study design, tools, …
- A e-learning resource dedicated to the rate of DRP and pharmacist intervention, using the classification published by Vo, T.H.;, et al. Clin. Pharm. Ther. 2018; 43: 240-248 (doi: 10.1111/jcpt.12642). The e-learning has been sent to all pharmacy students and their mentors one month before the study start and the completion was mandatory.
We completed the Method section as follow: “An e-learning resource was also sent to all pharmacy students and their mentors to help them train to score DRPs and apply standard-of-care PIs according to the French Society of Clinical Pharmacy. There was no other training” (line 80-83).
Line 84: eGFR is already abbreviated on line 47
Thank you, this is removed.
Line 85 and 86: Please provide references for aMDRD and CKD-EPI
We added the two following references:
For aMDRD: [15] Levey, A.S.; Bosch, J.P.; Lewis, J.B.; Greene, T.; Rogers, N.; Roth, D. A more accurate method to estimate glomerular filtration rate from serum creatinine: a new prediction equation. Modification of Diet in Renal Disease Study Group. Ann. Intern. Med. 1999; 130: 461-70.
For CKD-EPI: [16] Levey, A.S.; Stevens, LA.; Schmid, C.H.; Zhang, Y.L.; Castro 3rd, A.F.; Feldman, H.I.; Kusek, JW.; Eggers, P.; Van Lente, F.; Greene, T.; et al. A New Equation to Estimate Glomerular Filtration Rate. Ann. Intern. Med. 2009; 150: 604-613.
Figure 1: Please delete “a” in front of eGFR<60ml/min. Also, the unit for eGFR should be mL/min/1.73m2
This is deleted and corrected in the Figure 1 (lines 100-101).
Line 134: Was this satisfaction form in a Likert scale?
Yes, the satisfaction form has been build based on a Likert scale. We have added this information as follow: “[…] we asked community pharmacists to fill in a satisfaction form (using a Likert scale) about […]” (line 140).
Results:
Section 3.2: how many prescriptions were reviewed from 442 patients? This may be relevant information to determine the DRP prevalence in elderly patients with renal impairment.
There was n=4,508 prescribed medicines from n=442 patients. As previously described, the DRP rate was 2.2%. We have added this information in the Results section as follow: “Expert pharmacists reviewed all patients’ files (n=442) and detected 184 DRPs (4.1%) among 4,508 medicines, whereas community pharmacists detected n=99 also in 442 patients (2.2%)” (lines 185-187).
Also, were community pharmacists trained on DRP categories?
Yes, as previously explained the community pharmacists received an e-learning resource for training to score DRP and pharmacist intervention (according to the French Society of Clinical Pharmacy guidelines).
How many community pharmacists were involved in this study from 70 community pharmacies?
At least one community pharmacist (as a mentor) per community pharmacy was involved in this study.
Line 209: does 30 minutes including waiting for MD’s response after leaving the message?
Yes this includes all the time allocated to the GP contact and response. We completed the Method section as follow: “We also collected data on time spent for each patient’s record (less than 5 min, 5-10 min, 10-20 min, 20-30 min, and more than 30 min) including prescriber contact” (line 116-117).
Table 1:
Do you have any patient characteristics? For example, was DRP more common in advanced CKD stage? Or patients who are older than 70?
As request by Reviewer #1 we have added complementary information in the Results section regarding the mean of prescribed medicines per patient depending on age, gender and renal impairment stage: “In terms of patients prescribed medicine, there were no differences based on gender (10.5 ± 4.2 for male and 10.1±4.7 for female) or whether the patients were over 75 years old (10.3±4.6) or under (10.2±4.0). The number of prescribed medicines according to the stage of renal impairment was 9.5±4.4, 10.9±4.2, and 12.0±4.7 for stage 3A, stage 3B, and stage 4/5, respectively. Only 9.1% of patients were prescribed fewer than 5 medicines” (lines 178-182).
However we did not correlate these data with DRP detection.
Also, what do you mean by anti-gout “preparations”?
Anti-gout preparation is known as M04A in the ATC classification. This group include allopurinol, which was the most common medicine involved in the DRP detected.
The term anti-gout preparation has been used in the following articles, for example:
Bedouch, P.; Sylvoz, N.; Charpiat, B.; Juste, M.; Roubille, R.; Rose, F.-X.; Bosson J.-L.; Conort, O.; Allenet, B. Trends in pharmacists’s medication order review in French hospitals from 2006 to 2009: analysis of pharmacists’ interventions from the Act-IP website observatory. J. Clin. Pharm. Ther. 2015; 40: 32-40.
Laville, S.M.; Metzger, M.; Stengel, B.; Jacquelinet, C.; Combe, C.; Fouque, D.; Laville, M.; Frimat, L.; Ayav, C.; Speyer, E.; et al. Chronic Disease-Renal Epidemiology and Information Network (CKD-REIN) Study Collaborators. Evaluation of the adequacy of drug prescriptions in patients with chronic kidney disease: results from the CKD-REIN cohort. Br. J. Clin. Pharmacol. 2018, 84, 2811-2823.
How was five categories of DRP chosen? Or were you only able to determine five different categories within DRP? For instance, you did not see any DRP in failure to receive drug nor drug interaction.
All DRPs were described in the Result section. The study was designed to identify the DRP according to renal function, and not all DRPs performed by community pharmacists.
The aim of the study was to assess the impact of the systematic availability of eGFR to the community pharmacist on the DRP detection and management. The study was not design to detect all the DRPs in the patient with CKD. It is no surprising to highlight many DRP classified as contraindication or dose adjustment. The detection of DRP such as “failure to receive drug” or“drug interaction” seemed very unlikely in this context.
Please consider subcategorizing dosage problems into overdose and subtherapeutic dosage. This will be critical because majority (70%) of DRP is under dosage problems.
Probably because of the study objectives (eGFR-related DRPs), all (100%) of the DRPs detected by community pharmacists and by expert pharmacists were overdose.
Expert pharmacists found 4 DRP related to ACE inhibitors when 17 DRP were detected by CP. Of which, some intervention related to ACE inhibitors were considered as harmful to patients. By subcategorizing the dosage problem, it may explain the significance of DRPs.
As described above, there was only overdose-related DRPs.
Table 2:
I am not sure if this table adds value in this manuscript with these reasons:
- There were only 42 common DRP reported by community pharmacists and expert pharmacists.
- Thus, you will see strongly agreed on no DRP (because there aren’t many DRP)
- Some legends are not related to the table such as a definition of SE.
- Investigator should be replaced with community pharmacists to be consistent with Table 1
In accordance with your comment, we decided to remove the Table 2 and we report the significant results in the core text as follow: “There was strong agreement between CPs and expert pharmacists about the DRP category “no DRPs”, and agreement about the DRP categories “contraindication/non-conformity to guidelines” and “dosage problems”. We could find no statistically significant agreement in any other categories (higher than the expected agreement with a kappa statistic k = 0.168)” (lines 208-211).
Discussion:
- Line 229: I believe it should be greater or equal to 65 years old in the bracket.
This is corrected as follow: “of this population (≥65 years old)” (line, 234).
- Line 232-234: this sentence “it is always possible…” does not make sense.
We rephrased this part as follow: “Prescribers, typically the patient's GP, accepted half (52.5%) of the PIs suggested by CPs. In the French pilot study by Pourrat et al. the acceptance rate of pharmacist interventions was 33.3%, but the CPs only reported 18 interventions [11]. In our study, only 66 out of 99 interventions performed by CPs pharmacists were sent to prescribers, due to lack of GP availability and this perhaps could explain the difference” (lines 235-239).
- Line 276: I am confused. Analgesic DRP that was detected by experts (27%) were OTC drugs (NSAIDs)? If so, how did expert pharmacists were able to detect this DRP retrospectively when community pharmacists could not?
Regarding the study design, only prescribed medicines were included in the study. Our comment in the Discussion section tries to explain the difference between community and expert pharmacists. All the DRP regarding this pharmacological class were prescribed NSAIDs. We rephrased as follow: “NSAIDs are very common and can easily be overlooked. Moreover, our study only reviews prescribed medicines, and we could hypothesize that this DRP rate was underestimated in accordance with our study design” (line 288-290).
- Line 287: what type of dosage DRP did community pharmacist say about ACE inhibitor? Too high or too low? How would you know if the CPs did not know ACE inhibitors were used for renal protection?
As previously discussed, all the dosage DRP were overdose (according to eGFR, of course). In accordance to your question, we rephrased to temper the key message: “In our study, we could suggest that community pharmacists did not seem to be aware of this, as they reported almost all DRPs associated with ACE inhibitors (17% of DRP detected vs 2.2% for experts)” (lines 301-303).
- Line 291: what information makes you conclude that the training greatly improved DRP detection rate and intervention safety?
This is one hypothesis regarding the DRP detection rate difference between Community and Expert pharmacists. We also temper this key-message by replacing “greatly” by “probably” as follow: “[…] that training probably improves DRP detection rate and intervention safety” (line 305).
- It would be critical to discuss how your study was not able to detect DRP categories such as adverse drug events and untreated indication.
Even if our study could theoretically leads to the detection of all DRPs categories by Community pharmacists, the design clearly limits this. The systematic availability of eGFR has been proposed to encourage DRPs related to the renal safety of medicines before the DRP related to the specific CKD condition. Even if among the 442 patients recruited we did not collected comorbidities, we hypothesize that all the patients were not suffered from CKD. A large panel of DRP categories is more probable with a different design, for example focusing on CKD patients in an ambulatory setting such as day-hospital or a nephrologist conventional care consultation. This was performed by Belaiche et al. in a 6-month study targeting DRP in CKD patients [1]. They found n=142 DRP among 67 successive outpatients where the most frequent DRP was drug omission (31.7%) such as iron deficit in more than two-thirds of cases. This kind of DRP detection requests to have access to biological values other than eGFR that was not the case in our study. The other most frequent DRP were dosage problem (19.0%) and contraindication/non-conformity to guidelines (12.0%).
We have added discussion on this point in the Discussion section at lines 254-265 as well as the study by Belache et al. as a new reference (n°25).
[1] Belaiche, S.; Romanet, T.; Allenet, B.; Calop, J.; Zaoui, P. Identification of drug-related problems in ambulatory chronic kidney disease patients: a 6-month prospective study. J. Nephrol. 2012; 25: 782-788
References:
All references’ numbers are duplicated.
This is corrected.
Round 2
Reviewer 1 Report
I have reread the paper and the authors comments. I agree with these changes and feel that they have made significant changes and I approve of their changes